# Pirfenidone Alleviates Inflammation and Fibrosis of Acute Respiratory Distress Syndrome by Modulating the Transforming Growth Factor-β/Smad Signaling Pathway

**DOI:** 10.3390/ijms25158014

**Published:** 2024-07-23

**Authors:** Seung Sook Paik, Jeong Mi Lee, Il-Gyu Ko, Sae Rom Kim, Sung Wook Kang, Jin An, Jin Ah Kim, Dongyon Kim, Lakkyong Hwang, Jun-Jang Jin, Sang-Hoon Kim, Jun-Youl Cha, Cheon Woong Choi

**Affiliations:** 1Department of Medicine, Graduate School, Kyung Hee University, Seoul 02447, Republic of Korea; potshot@nate.com (S.S.P.); dy728@naver.com (D.K.); 2Department of Pulmonary, Allergy and Critical Care Medicine, Kyung Hee University Hospital at Gangdong, College of Medicine, Kyung Hee University, Seoul 05278, Republic of Korea; wjdal0916@naver.com (J.M.L.); kimsr0218@naver.com (S.R.K.); aikra@hanmail.net (S.W.K.); anjin7487@gmail.com (J.A.); 3Research Support Center, School of Medicine, Keimyung University, Deagu 42601, Republic of Korea; rhdlfrb@naver.com; 4Department of Nursing, Graduate School, Kyung Hee University, Seoul 02447, Republic of Korea; ginapon77@gmail.com; 5Team of Efficacy Evaluation, Orient Genia Inc., Seongnam 13201, Republic of Korea; lhwangphd@gmail.com (L.H.); threej09@hanmail.net (J.-J.J.); 6Department of Neurosurgery, Rutgers Robert Wood Johnson Medical School, The Stat University of New Jersey, Piscataway, NJ 08854, USA; sanghoon.kim1@rutgers.edu; 7Department of Sports and Martial Arts, Howon University, Gunsan 54058, Republic of Korea; roksfcha@howon.ac.kr

**Keywords:** acute respiratory distress syndrome, pirfenidone, inflammation, pulmonary fibrosis, lipopolysaccharide, bleomycin

## Abstract

Acute respiratory distress syndrome (ARDS) occurs as an acute onset condition, and patients present with diffuse alveolar damage, refractory hypoxemia, and non-cardiac pulmonary edema. ARDS progresses through an initial exudative phase, an inflammatory phase, and a final fibrotic phase. Pirfenidone, a powerful anti-fibrotic agent, is known as an agent that inhibits the progression of fibrosis in idiopathic pulmonary fibrosis. In this study, we studied the treatment efficiency of pirfenidone on lipopolysaccharide (LPS) and bleomycin-induced ARDS using rats. The ARDS rat model was created by the intratracheal administration of 3 mg/kg LPS of and 3 mg/kg of bleomycin dissolved in 0.2 mL of normal saline. The pirfenidone treatment group was administered 100 or 200 mg/kg of pirfenidone dissolved in 0.5 mL distilled water orally 10 times every 2 days for 20 days. The administration of LPS and bleomycin intratracheally increased lung injury scores and significantly produced pro-inflammatory cytokines. ARDS induction increased the expressions of transforming growth factor (TGF)-β1/Smad-2 signaling factors. Additionally, matrix metalloproteinase (MMP)-9/tissue inhibitor of metalloproteinase (TIMP)-1 imbalance occurred, resulting in enhanced fibrosis-related factors. Treatment with pirfenidone strongly suppressed the expressions of TGF-β1/Smad-2 signaling factors and improved the imbalance of MMP-9/TIMP-1 compared to the untreated group. These effects led to a decrease in fibrosis factors and pro-inflammatory cytokines, promoting the recovery of damaged lung tissue. These results of this study showed that pirfenidone administration suppressed inflammation and fibrosis in the ARDS animal model. Therefore, pirfenidone can be considered a new early treatment for ARDS.

## 1. Introduction

Acute respiratory distress syndrome (ARDS) is a disease that causes shock, sepsis, pneumonia, and ultimately leads to multiple organ failure. ARDS has an acute onset and presents with diffuse alveolar damage, refractory hypoxemia, and non-cardiac pulmonary edema [1]. Although airway management and protective ventilation technologies have advanced, the detailed mechanism of ARDS is not fully known, so the mortality rate of ARDS patients remains high.

The pathophysiology of ARDS is very complex, but inflammation and fibrosis are important causes of its pathogenesis. Clinically, ARDS follows an initial exudative, inflammatory phase and a final fibrotic phase in many patients [2]. Inflammatory cells accumulate in the lungs, and tumor necrosis factor-α (TNF-α), interleukin (IL)-1β, IL-6, and IL-8 are produced by these inflammatory cells [3]. At the onset of ARDS, the inflammation of pulmonary vascular endothelial cells and alveolar epithelial cells is promoted and lung tissue damage increases, leading to pulmonary fibrosis [4].

Many factors are involved in the formation of pulmonary fibrosis, such as extracellular matrix (ECM), cytokines, and inflammatory chemokines [4]. Transforming growth factor (TGF)-β exerts a critical role in the formation of pulmonary fibrosis by activating SMAD family member (Smad) signaling pathway [5]. Drugs that act on the TGF-β or Smad signaling pathways have potential as treatments for pulmonary fibrosis [6,7]. As another important mechanism of fibrosis progression, the balance of matrix metalloproteinases (MMPs) and tissue inhibitor metalloproteinases (TIMPs) is an important key to the formation of pulmonary fibrosis [4,8]. Fibroblasts derived from ARDS patients have been shown to induce impaired collagen remodeling, resulting in MMP/TIMP imbalance [9,10].

5-Methyl-1-phenyl-2-(1H)-pyridone (pirfenidone) is a strong antifibrotic agent that suppresses the progression of fibrosis in patients with idiopathic pulmonary fibrosis. Pirfenidone is an orally administered drug that is approved for the treatment of adults with mild to moderate idiopathic pulmonary fibrosis (IPF) in the EU and for the treatment of IPF in the USA [11]. Especially, pirfenidone attenuates profibrotic pathways. Alveolar epithelial cell (AEC) damage due to various factors induces AECs and endothelial cells to augment transforming growth factor-β (TGF-β) production [12]. In these cases, pirfenidone attenuates the transcription of procollagen, TGF-β, and platelet-derived growth factor and improves pulmonary fibrosis [13]. However, the exact mechanism by which pirfenidone inhibits pulmonary fibrosis is not yet fully known. In the current study, we evaluate the effectiveness of pirfenidone on the treatment of lipopolysaccharide (LPS) and bleomycin-induced ARDS.

For this study, bronchoalveolar lavage fluid (BALF) analysis was performed. Lung injury scores were calculated using hematoxylin and eosin (H&E) staining. Pulmonary fibrosis levels were detected by picrosirius red staining. Enzyme-linked immunosorbent assay (ELISA) was carried out to measure the levels of TNF-α, IL-1β, IL-6, IL-8, hydroxyproline, and connective tissue growth factor (CTGF). Western blotting was performed to detect the levels of TGF-β, Smad2, MMP-9, and TIMP-1.

## 2. Results

### 2.1. Cell Counting of BALF

The results of the BALF cell count are represented in Figure 1. The BALF cell number decreased upon ARDS induction (*p* < 0.05). However, pirfenidone treatment increased BALF cell number in the ARDS groups (*p* < 0.05). The results of the differential cell counts (eosinophils, neutrophils, and monocytes) are presented in Appendix A. Differential cell counts showed that the numbers of eosinophils and neutrophils increased with ARDS induction and pirfenidone treatment. However, the number of monocytes decreased in the ARDS group, whereas treatment with pirfenidone increased monocyte numbers.

### 2.2. Histological Analysis of Lung Injury Score and Pulmonary Fibrotic Level

Twenty-one days after the administration of LPS and bleomycin, intra-alveolar hemorrhage, interstitial edema, and inflammatory cell infiltration were observed in the alveolar lumen. Additionally, the lung tissue of the ARDS group showed the formation of fibrous bands or fibrous masses. These symptoms, such as inflammation and fibrosis, are indicators that ARDS has been triggered. Histological characteristics, lung injury scores, and pulmonary fibrosis levels are shown in Figure 2. These results mean that the lung injury score and pulmonary fibrosis level increased with ARDS induction (*p* < 0.05). However, pirfenidone treatment improved inflammatory cell infiltration, interstitial edema, and fibrous bands or masses, significantly reducing lung injury scores and lung fibrosis levels in the ARDS groups (*p* < 0.05).

### 2.3. Pro-Inflammatory Cytokine Concentrations

Figure 3 shows the concentration levels of TNF-α, IL-1β, IL-6, and IL-8 in lung tissue. The concentrations of TNF-α, IL-1β, IL-6, and IL-8 increased with ARDS induction (*p* < 0.05). But treatment of pirfenidone inhibited the concentrations of TNF-α, IL-1β, IL-6, and IL-8 in the ARDS groups (*p* < 0.05).

### 2.4. Expressions of TGF-β, Smad2, and Smad3

The expressions of TGF-β (25 kDa), Smad2 (55–60 kDa), and Smad3 (54 kDa) are shown in Figure 4. These results indicate that TGF-β, p-Smad2 (60 kDa), and p-Smad3 (54 kDa) expressions increased upon ARDS induction (*p* < 0.05). But the treatment of pirfenidone inhibited the expressions of TGF-β, p-Smad2, and p-Smad3 in the ARDS groups (*p* < 0.05).

### 2.5. Expressions of MMP-9 and TIMP-1

MMP-9 (92 kDa) and TIMP-1 (23 kDa) expressions are shown in Figure 5. These results indicate that the expression of MMP-9 was enhanced, and the expression of TIMP-1 was suppressed upon ARDS induction (*p* < 0.05). However, pirfenidone treatment suppressed the expression of MMP-9 and enhanced the expression of TIMP-1 in the ARDS groups (*p* < 0.05).

### 2.6. Expressions of Fibronectin, Hydroxyproline, and CTGF

Fibronectin (220 kDa) expression by Western blot and hydroxyproline and CTGF expressions by ELISA are presented in Figure 6. These results indicate that the expressions of fibronectin, hydroxyproline, and CTGF increased upon ARDS induction (*p* < 0.05). But the treatment of pirfenidone suppressed the expressions of fibronectin, hydroxyproline, and CTGF in the ARDS groups (*p* < 0.05).

## 3. Discussion

LPS contributes to acute lung injury, and LPS administration is used to produce lung injury in animal models. When LPS is in the bloodstream, it induces inflammation, which is an early clinical feature of lung injury [14]. Likewise, bleomycin-induced pulmonary fibrosis causes the fibrosis of lung tissue and is the most commonly used experimental method to evaluate the anti-fibrotic effect of drugs [15]. The ARDS model induced by LPS and bleomycin causes the early expression of inflammatory mediators, neutrophil or total leukocyte accumulation, and diffuse alveolar damage [3,16,17].

In the current study, the administration of LPS and bleomycin to rats though intratracheally resulted in alveolar capillary congestion, inflammatory cell infiltration, and pulmonary fibrosis. These changes increased lung injury score and lung fibrosis rate compared to control. The present results were similar to the results of previous studies demonstrating lung tissue damage caused by LPS and bleomycin [18,19]. The induction of ARDS causes the proliferation of inflammatory mediators that promote neutrophil accumulation in the lung microcirculation in the early stages [1].

In the current results, the injury effects of LPS and bleomycin to the lung may be due to the enhanced expressions of TNF-α, IL-1β, IL-6, and IL-8 in lung tissue. The number of BALF cells also increased. The results showed that the overproduction of pro-inflammatory cytokines exacerbated ARDS. Pirfenidone inhibits pro-inflammatory cytokines and chemokines and is associated with reducing inflammatory responses and T cell activity [20]. The current study showed that pirfenidone treatment substantially inhibited pro-inflammatory cytokine production. In previous studies, pirfenidone inhibited the expression of intracellular adhesion molecule (ICAM)-1 and enhanced the expression of anti-inflammatory cytokines such as IL-10 [21,22].

Pulmonary fibrosis appears in the final stages of ARDS progression and leads to decreased lung function and respiratory failure. When this condition occurs, effective treatment options are limited due to severe respiratory failure. TGF-β1 is a key cytokine in pulmonary fibrosis that participates in the Smad signaling pathway [23,24]. Smad2 is a major Smad family member and is phosphorylated upon TGF-β1 binding to form hetero-oligomeric complexes and regulate the transcription of specific genes [25]. The phosphorylation of Smad2 modulates collagen secretion, proliferation, and transformation and the excessive deposition of ECM in fibroblasts during pulmonary fibrosis [4,7]. Therefore, the modulation of the TGF-β1/Smad signaling pathway is important for improving lung fibrosis in ARDS as it is involved in inflammation, mesenchymal transition, and ECM deposition [26,27]. In the present results, TGF-β1, p-Smad2, and p-Smad3 levels were clearly enhanced by intratracheal LPS and bleomycin injection, and treatment with pirfenidone clearly suppressed LPS/bleomycin-induced TGF-β1, p-Smad2, and p-Smad3 expressions. The results of this experiment are consistent with those of other researchers, showing that pirfenidone treatment inhibits TGF-β1 expression [13,24].

ECM and collagen deposition play a critical role for the formation of pulmonary fibrosis in ARDS. ECM is mostly degraded by MMPs, and the degradation activity of MMPs is modulated by TIMPs [4]. In patients with ARDS, the excessive production of MMP-9 can destroy the basement membrane, allow fibroblasts to invade the alveolar space, and lead to pulmonary fibrosis [28,29]. MMP-9 is constitutively suppressed by TIMP-1, such that an imbalance between MMP-9 and TIMP-1 plays a role in the pathogenesis of ARDS [28]. In the present study, we investigated MMP-9 and TIMP-1 levels in the lung tissue of LPS- and bleomycin-induced ARDS rats. Through ARDS induction, the expression of MMP-9 was upregulated, while TIMP-1 expression decreased. Pirfenidone inhibits the expression of MMP-9 and TIMP1 directly or by reducing the synthesis of TGF-β and downstream mediators. Pirfenidone may also reduce TGF-β activation by MMPs [12]. Currently, the balance of MMP-9 and TIMP-1 was regulated by pirfenidone treatment in LPS- and bleomycin-induced ARDS rats. The present results are consistent with previous reports showing that pirfenidone treatment modulates MMP-9 and TIMP-1 expression [30,31].

Fibronectin, which acts a critical role in cell adhesion, migration, growth, and differentiation, mediates various cellular interactions with the ECM. Hydroxyproline is a neutral heterocyclic protein amino acid. It is present in collagen and is commonly found in many gelatin products. Hydroxyproline is primarily used as a diagnostic marker for bone turnover and liver fibrosis. CTGF plays many roles in cell adhesion, migration, proliferation, angiogenesis, skeletal development, and tissue wound repair and is implicated in the determination of fibrotic diseases and several forms of cancer [32,33]. In the present results, ARDS induction enhanced the expressions of fibronectin, hydroxyproline, and CTGF, whereas pirfenidone treatment decreased these expressions in lung tissue. The therapeutic effect of pirfenidone on fibrotic injury is thought to be due to its ability to modulate MMP-9 and TIMP-1 expression.

The current study showed that pirfenidone treatment suppressed inflammation and fibrosis, suppressed lung injury score, and improved immunological imbalance in an ARDS rat model. Pirfenidone has been reported to have side effects involving gastrointestinal and skin conditions [34,35]. However, there are no known side effects that are severe enough to consider stopping taking the drug. Therefore, pirfenidone can be considered as a new early treatment for ARDS.

## 4. Materials and Methods

### 4.1. Animal Classification

For the experiment, 40 male adult Sprague Dawley rats weighing 200 ± 5 g (8 weeks old), were obtained from Orient Co. (Seongnam, Republic of Korea). The rats were randomly classified into four groups (n = 10 for each group): control group; ARDS-induced group; ARDS-induced and 100 mg/kg pirfenidone-treated group; and ARDS-induced and 200 mg/kg pirfenidone-treated group. The experimental procedures were obtained following approval number KHUASP(SE)-18-036 by the Institutional Animal Care and Use Committee of Kyung Hee University.

### 4.2. ARDS Rat Model

The ARDS rat model was created in the following way [16,17,18]. After anesthesia by injection of 10 mg/kg Zoletil 50^®^ (Vibac Laboratories, Carros, France) intraperitoneally, 3 mg/kg of LPS (Sigma Aldrich Co., St. Louis, MO, USA) and 3 mg/kg of bleomycin (Tokyo Chemical Industry, Tokyo, Japan) in 0.2 mL saline were injected intratracheally to cause lung damage. Treatment for the control rats included intratracheal injection of an equal volume of saline. On the 1st day after ARDS induction, 100 mg/kg or 200 mg/kg of pirfenidone (Kolon Pharmaceutical, Gwacheon, Republic of Korea) was dissolved in 0.5 of mL of distilled water and administered through the mouth to rats in the pirfenidone-treated groups, once every two days for 20 days. Rats in the drug-free groups received 0.5 mL of distilled water orally without pirfenidone on the same schedule. We selected the pirfenidone concentration that had been found to result in high efficacy and survival rates through a preliminary experiment.

### 4.3. Bronchoalveolar Lavage Fluid and Lung Tissue Slice

The rats were sacrificed 21 days after the administration of LPS and bleomycin. After anesthesia by injection of 10 mg/kg of Zoletil 50^®^ (Vibac Laboratories) intraperitoneally, the trachea was separated through an incision, and a small-bore tube was inserted and secured into the trachea. Next, phosphate-buffered saline (pH 7.2) was slowly injected into the lungs, and BALF was withdrawn through the inserted tube. The right lobe of the lung was then removed. The collected lung tissues were fixed in 4% paraformaldehyde, dehydrated with graded ethanol, treated with xylene, and embedded by paraffin infiltration. Coronal sections of 5 μm thickness were created using a paraffin microtome (Thermo Co., Cheshire, UK), and these sections were placed on coated slides. Slides were dried on a hot plate overnight at 37 °C. Six slice sections were created from each lung sample.

### 4.4. BALF Cell Counting

BALF cell counting was carried out in the following way [3,36]. BALF cell suspension was diluted at 1:20 with trypan blue. After loading into the hemocytometer chamber and settling the cells, cells were counted in four corner squares. Cell differential counts were performed on cytospin preparations (Shadon, Pittsburgh, PA, USA), and the cells were stained with Diff-Quick (Fisher Scientific; Fair Lawn, NJ, USA). Total leukocytes were counted using light microscopy (Olympus, Tokyo, Japan).

Furthermore, differential counts of eosinophils, neutrophils, and monocytes were determined on cytospin smears of BAL samples from individual rat stained with Diff-Quick and identified by standard morphological criteria after counting cells.

### 4.5. Pro-Inflammatory Cytokines and Fibrosis-Related Factors

ELISA was used to detect the concentration levels of pro-inflammatory cytokines (TNF-α, IL-1β, IL-6, and IL-8) and fibrosis-related factors (hydroxyproline and CTGF) in the lung tissue. The assay was conducted according to the manufacturer’s protocol (Abcam, Cambridge, UK) using an enzyme immunoassay kit [37,38].

### 4.6. H&E Staining and Lung Injury Score

H&E staining was conducted in the following way [24,39]. The slides were dipped in Mayer’s hematoxylin (DAKO, Glostrup, Denmark) for 30 s, rinsed with water until the slides became transparent, dipped in eosin (Sigma Aldrich) for 10 s, and rinsed again with water. After drying the slides at room temperature, they were soaked twice in 95% ethanol, 100% ethanol, 50% ethanol, 50% xylene, and 100% xylene solution, and coverslipped with Permount^®^ (Fisher Scientific, Waltham, MA, USA).

Lung injury scores were evaluated in the following way [3]. Images of H&E-stained slides were acquired using Image-Pro^®^ plus computer-assisted image analysis system (Media Cyberbetics Inc., Silver Spring, MD, USA) attached to a light microscope (Olympus, Tokyo, Japan). The sections were assessed for alveolar capillary congestion, hemorrhage, or infiltration or the aggregation of inflammatory cells within the airspace, and alveolar wall/hyaline membrane formation thickness, with each feature scored from 0 to 3 (0 = absent; 1 = mild; 2 = moderate; 3 = prominent).

### 4.7. Picrosirius Red Staining

Picrosirius red staining was conducted in the following way [40]. The sections were stained with a picrosirius red staining kit (Abcam) as per the manufacturer’s instructions and coverslipped using Permount^®^ (Fisher Scientific). Images of picrosirius red-stained slides were acquired using Image-Pro^®^ plus a computer-assisted image analysis system (Media Cyberbetics Inc.) attached to an optical microscope (Olympus). Five fields of view were randomly selected from each sample and imaged using the following method [41]. The areas of collagen (red) and muscle tissue (yellow) were quantified. The integrated optical density of collagen versus muscle was calculated as a percentage and compared between groups.

### 4.8. Western Blotting

Western blotting was performed as follows [3,42]. After lung tissues were homogenized using lysis buffer, they were centrifuged at 14,000 rpm for 30 min. After the detection of protein contents by a Bio-Rad colorimetric protein assay kit (Bio-Rad, Hercules, CA, USA), the separation of 30 μg of protein in SDS-polyacrylamide gels followed, and next they were transferred onto a nitrocellulose membrane. Table 1 is the list of primary and secondary antibodies used for this experiment. All experiment steps were carried out at room temperature except for membrane transfer that was conducted at 4 °C using a cold pack and prechilled buffer. Band detection was performed by the enhanced chemiluminescence (ECL) detection kit (Santa Cruz Biotechnology, Dallas, TX, USA), and then it was detected by Molecular Analyst^TM^ version 1.4.1 (Bio-Rad).

### 4.9. Statistical Analysis

One-way analysis of variance with Duncan’s post hoc test by SPSS software (Version 23.0, IBM Co., Armonk, NY, USA) was carried out for statistical analysis. *p* < 0.05 was set to indicate statistical significance.

## Figures and Tables

**Figure 1 ijms-25-08014-f001:**
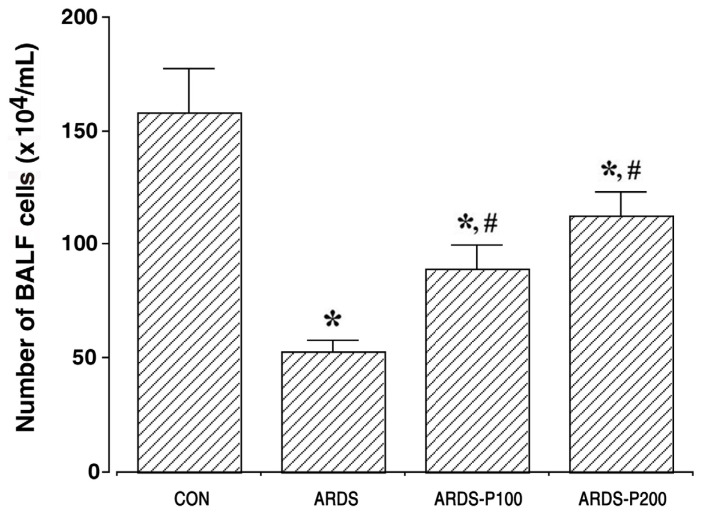
Cell counting of bronchoalveolar lavage fluid. * indicates *p* < 0.05 compared to the control group. # indicates *p* < 0.05 compared to the ARDS-induced group. CON, control group; ARDS, acute respiratory distress syndrome-induced group; ARDS-P100, acute respiratory distress syndrome-induced and 100 mg/kg pirfenidone-treated group; ARDS-P200, acute respiratory distress syndrome-induced and 200 mg/kg pirfenidone-treated group.

**Figure 2 ijms-25-08014-f002:**
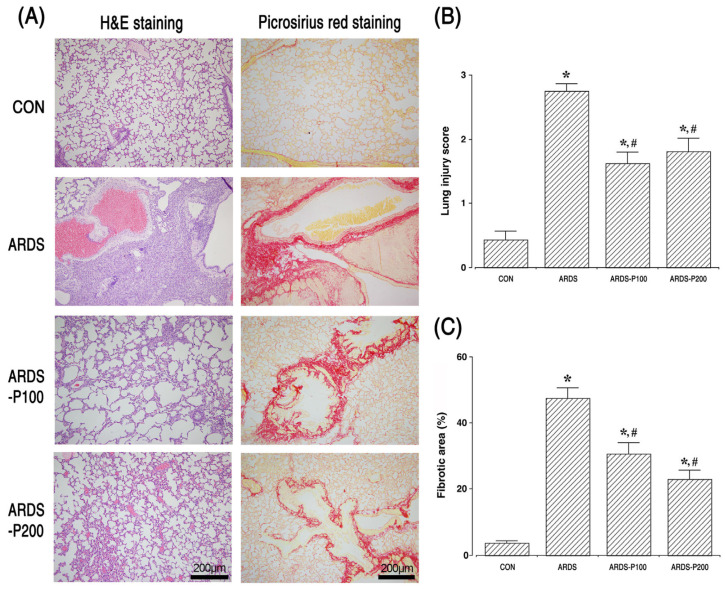
Histological analysis, lung injury score, and pulmonary fibrotic level. (**A**) Hematoxylin and eosin (H&E) staining and picrosirius red staining. Scale bar is 200 µm. (**B**) Lung injury score. (**C**) Fibrotic area. * indicates *p* < 0.05 compared to the control group. # indicates *p* < 0.05 compared to the ARDS-induced group. CON, control group; ARDS, acute respiratory distress syndrome-induced group; ARDS-P100, acute respiratory distress syndrome-induced and 100 mg/kg pirfenidone-treated group; ARDS-P200, acute respiratory distress syndrome-induced and 200 mg/kg pirfenidone-treated group.

**Figure 3 ijms-25-08014-f003:**
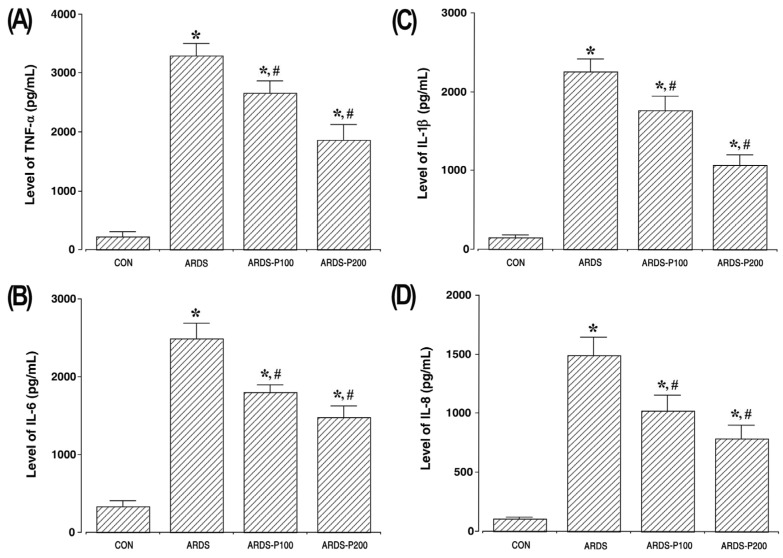
Pro-inflammatory cytokine concentrations. (**A**) The level of tumor necrosis factor (TNF)-α in the lung tissue. (**B**) The level of interleukin (IL)-6 in the lung tissue. (**C**) The level of IL-1β in the lung tissue. (**D**) The level of IL-8 in the lung tissue. * indicates *p* < 0.05 compared to the control group. # indicates *p* < 0.05 compared to the ARDS-induced group. CON, control group; ARDS, acute respiratory distress syndrome-induced group; ARDS-P100, acute respiratory distress syndrome-induced and 100 mg/kg pirfenidone-treated group; ARDS-P200, acute respiratory distress syndrome-induced and 200 mg/kg pirfenidone-treated group.

**Figure 4 ijms-25-08014-f004:**
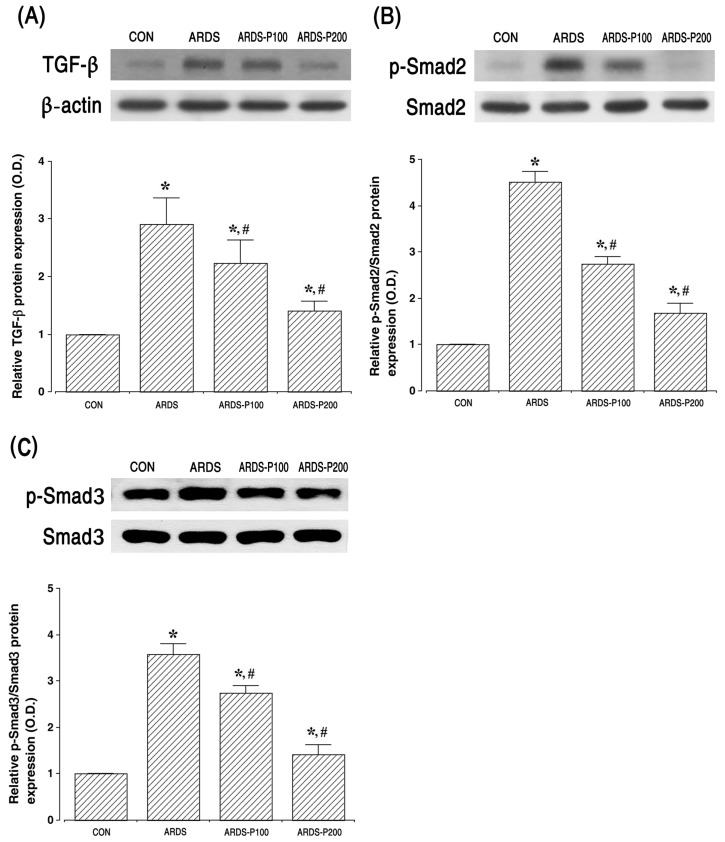
Expressions of transforming growth factor (TGF)-β, SMAD family member 2 (Smad2), and Smad3. (**A**) The relative TGF-β expression in the lung tissue. (**B**) The relative ratio of phosphorylated Smad2 (p-Smad2) to Smad2 in the lung tissue. (**C**) The relative ratio of phosphorylated Smad3 (p-Smad3) to Smad3 in the lung tissue. * indicates *p* < 0.05 compared to the control group. # indicates *p* < 0.05 compared to the ARDS-induced group. CON, control group; ARDS, acute respiratory distress syndrome-induced group; ARDS-P100, acute respiratory distress syndrome-induced and 100 mg/kg pirfenidone-treated group; ARDS-P200, acute respiratory distress syndrome-induced and 200 mg/kg pirfenidone-treated group.

**Figure 5 ijms-25-08014-f005:**
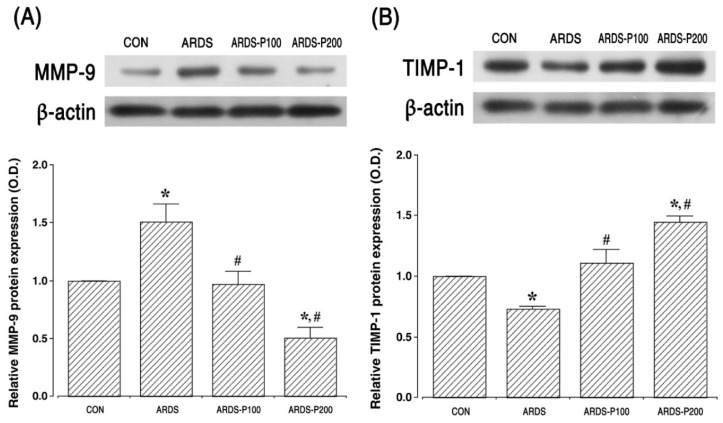
Expressions of matrix metalloproteinase (MMP)-9 and tissue inhibitor metalloproteinase (TIMP)-1. (**A**) The relative MMP-9 expression in the lung tissue. (**B**) The relative TIMP-1 expression in the lung tissue. * indicates *p* < 0.05 compared to the control group. # indicates *p* < 0.05 compared to the ARDS-induced group. CON, control group; ARDS, acute respiratory distress syndrome-induced group; ARDS-P100, acute respiratory distress syndrome-induced and 100 mg/kg pirfenidone-treated group; ARDS-P200, acute respiratory distress syndrome-induced and 200 mg/kg pirfenidone-treated group.

**Figure 6 ijms-25-08014-f006:**
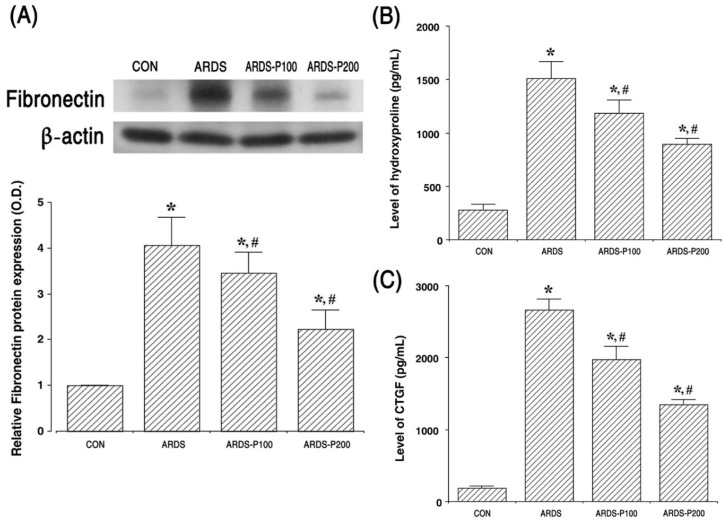
Expression of fibronectin and concentrations of hydroxyproline and connective tissue growth factor (CTGF). (**A**) The relative fibronectin expression in the lung tissue. (**B**) The level of hydroxyproline in the lung tissue. (**C**) The level of CTGF in the lung tissue. * indicates *p* < 0.05 compared to the control group. # indicates *p* < 0.05 compared to the ARDS-induced group. CON, control group; ARDS, acute respiratory distress syndrome-induced group; ARDS-P100, acute respiratory distress syndrome-induced and 100 mg/kg pirfenidone-treated group; ARDS-P200, acute respiratory distress syndrome-induced and 200 mg/kg pirfenidone-treated group.

**Table 1 ijms-25-08014-t001:** Primary and secondary antibodies in Western blot.

Classification	Items	Source	Titer	Company
Primary antibody	Smad2, p-Smad2, Smad3, p-Smad3 MMP-9, TIMP-1, Fibronectin, β-actin	Anti-mouse	1:1000	Cell Signaling Technology, Danvers, DA, USA Santa Cruz Biotechnology, Dallas, TX, USA
	TGF-β1	Anti-rabbit	1:1000	Cell Signaling Technology, Danvers, DA, USA
Secondary antibody	HRP-conjugated IgG	Mouse Rabbit	1:2000	Vector Laboratories, Newark, CA, USA

## Data Availability

All data generated in this study are included in this manuscript.

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
