# Peer review of "Pirfenidone Alleviates Inflammation and Fibrosis of Acute Respiratory Distress Syndrome by Modulating the Transforming Growth Factor-β/Smad Signaling Pathway"

_ijms, 2024, doi:10.3390/ijms25158014_

Round 1

Reviewer 1 Report

Comments and Suggestions for Authors

I have thoroughly reviewed the manuscript titled "Pirfenidone alleviates inflammation and fibrosis of acute respiratory distress syndrome by modulating the transforming growth factor-β/Smad signaling pathway”. The manuscript is written well, however, I have several comments or suggestion that needs to be answered

·        I will suggest to add a more detailed literature review on pirfenidone's known mechanisms of action and its current clinical use in the introduction section.

·        I will suggest to provide more detailed information on the experimental design, specifically the rationale behind choosing 100 mg/kg and 200 mg/kg doses of pirfenidone. How were these doses determined, and are they reflective of clinically relevant dosages?

·        I will suggest to add magnification used to capture the histology images.

·        I will suggest to add specific molecular weight of the protein used for western blot.

·        The method for bronchoalveolar lavage fluid (BALF) cell counting lacks specificity. The manuscript should detail how different cell types (e.g., neutrophils, macrophages) were identified and counted.

·        The discussion on MMP-9 and TIMP-1 imbalance is crucial but lacks depth. The manuscript should explore the underlying mechanisms that lead to this imbalance and how pirfenidone specifically alters these pathways.

·        The manuscript does not address the potential side effects of pirfenidone treatment. Including a discussion on any observed side effects or referencing studies on pirfenidone's safety profile would provide a more comprehensive overview.

·        I will suggest to add a graphical abstract

·        There are several typographical errors, should check carefully

Author Response

Answers to Reviewers’ Comments

Journal name: International Journal of Molecular Sciences
Manuscript ID: ijms-3054832
Type of manuscript: Article
Title: Pirfenidone alleviates inflammation and fibrosis of acute respiratory distress syndrome by modulating the transforming growth factor-β/Smad signaling pathway
Authors: Seung Sook Paik, Jeong Mi Lee, Il-Gyu Ko, Sae Rom Kim, Sung Wook Kang, Jin An, Jin Ah Kim, Dongyon Kim, Lakkyong Hwang, Jun-Jang Jin, Sang-Hoon Kim, Jun-Youl Cha, Cheon Woong Choi*

We sincerely appreciate for your kind advice and comments to our manuscript. We revised the manuscript according to the reviewer’s comments. We added new experimental data, and modifications were expressed in red.

Reviewer -1

I have thoroughly reviewed the manuscript titled "Pirfenidone alleviates inflammation and fibrosis of acute respiratory distress syndrome by modulating the transforming growth factor-β/Smad signaling pathway”. The manuscript is written well, however, I have several comments or suggestion that needs to be answered.

Q1. I will suggest to add a more detailed literature review on pirfenidone's known mechanisms of action and its current clinical use in the introduction section.

Author answer: First all, thank you for your kind advice and comments. According to reviewer’s comment, the modified sentence and adding references were inserted in manuscript.

  • Following sentence were added to the Introduction part

5-Methyl-1-phenyl-2-(1H)-pyridone (pirfenidone) is a strong antifibrotic agent that suppresses progression of fibrosis in the patients with idiopathic pulmonary fibrosis. Pirfenidone is an orally administered that is approved for the treatment of adults with mild to moderate idiopathic pulmonary fibrosis (IPF) in the EU, and for the treatment of IPF in the USA [11]. Especially, pirfenidone attenuates profibrotic pathways. Alveolar epithelial cell (AEC) damage due to various factors induces AECs and endothelial cells to augment transforming growth factor-β (TGF-β) production [12]. In these cases, pirfenidone attenuates transcription of procollagen, TGF-β, and platelet-derived growth factor and improves pulmonary fibrosis [13]. However, the exact mechanism by which pirfenidone inhibits pulmonary fibrosis is not yet fully known. In the current study, we evaluated the effectiveness of pirfenidone on the treatment of lipopolysaccharide (LPS) and bleomycin-induced ARDS.

  • Following reference were added to the reference part
  1. Kim, E.S.; Keating, G.M. Pirfenidone: A Review of Its Use in Idiopathic Pul-monary Fibrosis. Drugs. 2015, 219-230. http://doi.org/10.1007/s40265-015-0350-9
  2. Saleela, M.; Ruwanpura, B.J.T.; Philip, G.; Bardin. Pirfenidone: Molecular Mechanisms and Potential Clinical Applications in Lung Disease. Am J Respir Cell Mol Biol. 2020, 62, 413-422. http://doi.org/ 10.1165/rcmb.2019-0328TR

Q2. I will suggest to provide more detailed information on the experimental design, specifically the rationale behind choosing 100 mg/kg and 200 mg/kg doses of pirfenidone. How were these doses determined, and are they reflective of clinically relevant dosages?

Author answer: We conducted a preliminary experiment before conducting this study. The purpose of the preliminary experiment was to select an animal model and effective concentration. The effective concentration was constructed by considering the dose of 200 mg in clinical use. As a result of preliminary experiments, the condition of the experimental animals (in non-ARDS model) was very poor depending on whether pirfenidone was administered, and some deaths were observed even at high doses. And after bleomycin+LPS administrations, many deaths occurred in the high-dose pirfenidone group (Preliminary experiment result-1).

Preliminary experiment result-1. Changes in survival rate with pirfenidone administration. (A) Changes in survival rates following pirfenidone administration in normal conditions. (B) Changes in survival rates following pirfenidone administration in ARDS conditions. (□) Control group, (▲) 10mg/kg pirfenidone-treated group, (○) 50mg/kg pirfenidone-treated group, (■) 100mg/kg pirfenidone-treated group, (∆) 200mg/kg pirfenidone-treated group, (●) 400mg/kg pirfenidone-treated group, (×) 800mg/kg pirfenidone-treated group.

In addition, preliminary experimental results showed that the anti-inflammatory and anti-fibrotic results of each dose of pirfenidone were major factors in dose selection (Preliminary experiment result-2). This was a very important issue in experimental design.

Preliminary experiment result-2. Changes in TNF-α and CTGF expression by dose-dependent pirfenidone administration in ARDS model. (A) Level of TNF-α, (B) Level of CTGF. (CON) Control group, (A-P10) ARDS-induced and 10mg/kg pirfenidone-treated group, (A-P50) ARDS-induced and 50mg/kg pirfenidone-treated group, (A-P100) ARDS-induced and 100mg/kg pirfenidone-treated group, (A-P200) ARDS-induced and 200mg/kg pirfenidone-treated group, (A-P400) ARDS-induced and 400mg/kg pirfenidone-treated group, (A-P800) ARDS-induced and 800mg/kg pirfenidone-treated group. *Shows P < 0.05 compared to the control group, # Shows P < 0.05 compared to the ARDS-induced and 10mg/kg pirfenidone-treated group.

Therefore, this study selected the optimal concentration by considering the concentration in clinical use among the effective concentrations that are easy to conduct research. According to reviewer’s comment, a new sentence has been inserted for dosages in this study in material & methods part.

  • Following sentence were inserted in the Material & methods part

We selected the pirfenidone concentration that found to considering high efficacy and survival rate through preliminary experiment.

Q3. I will suggest to add magnification used to capture the histology images.

Author answer: According to reviewer comment, we added magnification to the histological image in the Result part.

  • Following figure were inserted in the result part.

Q4. I will suggest to add specific molecular weight of the protein used for western blot.

Author answer: According to reviewer comment, we add specific molecular weight of the protein used for western blot in the Result parts.

  • Following sentence were inserted in the Results part

The expressions of TGF-β (25 kDa), Smad2 (55-60 kDa), and Smad3 (54 kDa) are shown in Fig. 4. These results indicate that TGF-β, p-Smad2 (60 kDa), and p-Smad3 (54 kDa) expressions were increased upon ARDS induction (P < 0.05).

MMP-9 (92 kDa) and TIMP-1 (23 kDa) expressions are shown in Fig. 5.

Fibronectin (220 kDa) expression by western blot and hydroxyproline and CTGF expressions by ELISA are presented in Fig. 6

Q5. The method for bronchoalveolar lavage fluid (BALF) cell counting lacks specificity. The manuscript should detail how different cell types (e.g., neutrophils, macrophages) were identified and counted.

Author answer: According to the reviewer's opinion, modifications to cell counts and references addition of the BALF method were inserted into the manuscript.

  • Following sentence were inserted in the Material & methods part

4.4. BALF cell counting

BALF cell counting was done in the following way [3,36]. BALF cell suspension was diluted at 1:20 with trypan blue. After loading into the hemocytometer chamber and settling the cells, cells were counted in four corner squares. Cell differential counts were per-formed on cytospin preparations (Shadon, Pittsburgh, PA, USA), and the cells were stained with Diff-Quick (Fisher Scientific; NJ, USA). Total leukocytes were counted using light microscopy (Olympus, Tokyo, Japan).

  • Following reference were added to the reference part
  1. Kwon, do, Y.; Kim, H.M.; Kim, E.; Lim, Y.M.; Kim, P.; Choi, K.; Kwon, J.T. Acute pulmonary toxicity and inflammation induced by combined exposure to didecyldimethylammonium chloride and ethylene glycol in rats. J Toxicol Sci. 2016, 41, 17-24. http://doi.org/10.2131/jts.41.17.

  • Following reference were deleted to the reference part
  1. Zeidler-Erdely, P.C.; Antonini, J.M.; Meighan, T.G.; Young, S.H.; Eye, T.J.; Hammer, M.A.; Erdely, A. Comparison of cell counting methods in rodent pulmonary toxicity studies: automated and manual protocols and considera-tions for experimental design. Inhal Toxicol. 2016, 28, 410-420. http://doi.org/10.1080/08958378.2016.1189985

 Q6. The discussion on MMP-9 and TIMP-1 imbalance is crucial but lacks depth. The manuscript should explore the underlying mechanisms that lead to this imbalance and how pirfenidone specifically alters these pathways.

Author answer: We agree with the reviewer’s opinion. Pirfenidone attenuates pro-fibrotic pathways. AEC damage due to environmental or other factors induces AECs and endothelial cells to augment TGF-β production.

Four key processes are activated: proliferation of fibroblasts, trans differentiation of fibroblasts into myofibroblasts, collagen synthesis and fibronectin production, and excess ECM production. Targets of pirfenidone include TGF-β itself and TGF-β–induced downstream mediators and products such as SMAD3; α-SMA; tenascin-c; fibronectin; collagen type I, II and III; and the collagen-specific chaperone HSP47. Pirfenidone also regulates additional growth factors such as PDGF and bFGF, thereby modulating collagen production. Pirfenidone also inhibits expression of MMP-9, TIMP1, and MMP-2 (yellow boxes) directly or by reducing the synthesis of TGF-β and downstream mediators. Pirfenidone may also reduce TGF-β activation by MMPs (Saleela et al., 2020). According to reviewer’s comment, the modified sentence and adding references were inserted in manuscript.

  • Following sentence were inserted in the Discussion part

By ARDS induction, the expression of MMP-9 was upregulated, whereas TIMP-1 expression was decreased. Pirfenidone inhibits expression of MMP-9 and TIMP1 diectly or by reducing the synthesis of TGF-β and downstream mediators. Pirfenidone may also reduce TGF-β activation by MMPs [12]. Currently, the balance of MMP-9 and TIMP-1 was regulated by pirfenidone treatment in LPS and bleomycin-induced ARDS rats.

  • Following reference were added to the reference part
  1. Saleela, M.; Ruwanpura, B.J.T.; Philip, G.; Bardin. Pirfenidone: Molecular Mechanisms and Potential Clinical Applications in Lung Disease. Am J Respir Cell Mol Biol. 2020, 62, 413-422. http://doi.org/ 10.1165/rcmb.2019-0328TR

Q7. The manuscript does not address the potential side effects of pirfenidone treatment. Including a discussion on any observed side effects or referencing studies on pirfenidone's safety profile would provide a more comprehensive overview.

Author answer: The most common side effects of pirfenidone are gastrointestinal disorders and skin diseases. If symptoms are not severe, use drugs that regulate gastrointestinal motility, sunscreen, etc. Occasionally, if symptoms are severe, the drug may be discontinued, but stopping the drug is not common. According to reviewer’s comment, a new sentence and reference has been inserted in the manuscript.

  • Following sentence were inserted in the Discussion part

Pirfenidone has been reported to have side effects of some gastrointestinal and skin diseases [32,33]. However, there are no known side effects that are severe enough to consider stopping taking the drug. Therefore, pirfenidone can be considered as a new early treatment for ARDS.

  • Following reference were inserted in the reference part
  1. Cottin, V.; Maher, T. Long-term clinical and real-world experience with pirfenidone in the treatment of idiopathic pulmonary fibrosis. Eur Respir Rev. 2015, 24, 58-64. http://doi.org/10.1183/09059180.00011514.

  1. Hughes, G.; Toellner, H.; Morris, H.; Leonard, C.; Chaudhuri, N. Real World Experiences: Pirfenidone and Nintedanib are Effective and Well Tolerated Treatments for Idiopathic Pulmonary Fibrosis. J Clin Med. 2016, 5, 78. http://doi.org/10.3390/jcm5090078.

Q8. I will suggest to add a graphical abstract

Author answer: According to reviewer’s comment, a new graphic abstract has been inserted in the manuscript.

  • Following figure were inserted in the manuscript

Q9. There are several typographical errors, should check carefully

Author answer: As instructed by the reviewer's, we carefully checked and modified the several typographical errors.

Reviewer 2 Report

Comments and Suggestions for Authors

In the manuscript entitled "Pirfenidone alleviates inflammation and fibrosis of acute respiratory distress syndrome by modulating the transforming growth factor-β/Smad signaling pathway", Seung Sook Paik et al. describe the results obtained in a murine model of acute respiratory distress syndrome (ARDS) induced by LPS and bleomycin, evaluating the therapeutic effect of pirfenidone. The results suggest that pirfenidone alleviates ARDS-associated lung inflammation and fibrosis, reducing histological alterations in the lung and inflammatory cytokines, including TNF-α, IL-1β, IL-6 and IL-8, by regulating the TGF-β/Smad pathway.

Overall, the study presents significant contributions to the field. However, I have some minor observations regarding the submitted manuscript.

Figure 1: The authors show the results of total cell count in BALF, indicating that the total number of cells decreases after induction of ARDS and that treatment with pirfenidone recovers the number of cells in BALF. However, some references cited in the manuscript suggest that induction of ARDS with LPS promotes an increase of inflammatory cells in BALF, not a decrease. How do the authors justify these contradictory results? It would be beneficial to include a more detailed discussion in this regard or perform additional experiments to clarify this discrepancy.

The authors highlight that pirfenidone alleviates ARDS inflammation and fibrosis by modulating the TGF-β/Smad signaling pathway. Although they measure TGF-β1, Smad2 and P-Smad2 levels, binding of TGF-β1 to its receptors not only activates and phosphorylates Smad2, but also Smad3 and P-Smad3. It would be important for the authors to evaluate the levels of Smad3 and P-Smad3 to provide a more complete picture of the mechanism of action of pirfenidone in modulating this signaling pathway.

Author Response

Answers to Reviewers’ Comments

Journal name: International Journal of Molecular Sciences
Manuscript ID: ijms-3054832
Type of manuscript: Article
Title: Pirfenidone alleviates inflammation and fibrosis of acute respiratory distress syndrome by modulating the transforming growth factor-β/Smad signaling pathway
Authors: Seung Sook Paik, Jeong Mi Lee, Il-Gyu Ko, Sae Rom Kim, Sung Wook Kang, Jin An, Jin Ah Kim, Dongyon Kim, Lakkyong Hwang, Jun-Jang Jin, Sang-Hoon Kim, Jun-Youl Cha, Cheon Woong Choi*

We sincerely appreciate for your kind advice and comments to our manuscript. We revised the manuscript according to the reviewer’s comments. We added new experimental data, and modifications were expressed in red.

In the manuscript entitled "Pirfenidone alleviates inflammation and fibrosis of acute respiratory distress syndrome by modulating the transforming growth factor-β/Smad signaling pathway", Seung Sook Paik et al. describe the results obtained in a murine model of acute respiratory distress syndrome (ARDS) induced by LPS and bleomycin, evaluating the therapeutic effect of pirfenidone. The results suggest that pirfenidone alleviates ARDS-associated lung inflammation and fibrosis, reducing histological alterations in the lung and inflammatory cytokines, including TNF-α, IL-1β, IL-6 and IL-8, by regulating the TGF-β/Smad pathway. Overall, the study presents significant contributions to the field. However, I have some minor observations regarding the submitted manuscript.

Q1. Figure 1: The authors show the results of total cell count in BALF, indicating that the total number of cells decreases after induction of ARDS and that treatment with pirfenidone recovers the number of cells in BALF. However, some references cited in the manuscript suggest that induction of ARDS with LPS promotes an increase of inflammatory cells in BALF, not a decrease. How do the authors justify these contradictory results? It would be beneficial to include a more detailed discussion in this regard or perform additional experiments to clarify this discrepancy.

Author answer: First all, thank you for your kind advice and comments. We admit that our choice of reference was wrong. There are very many differences in design between the studies cited and our studies. In particular, Shah's study (2014) showed that the increase in BALF peaked on the 3rd day of LPS administration, but decreased on the 7th day. Our study administers drugs for 21 days after inducing the ARDS model. Therefore, there is a difference from excessive increase in the acute phase. We have been studying various types of respiratory diseases. The BALF analysis results showed a similar trend in this study. We hope that the reviewer will consider these points. According to reviewer’s comment, the modified references were inserted in manuscript.

  • Following reference were inserted in the manuscript

The ARDS model induced by LPS and bleomycin causes early expression of inflammatory mediators, neutrophil or total leukocyte accumulation, and diffuse alveolar damage [3,14,15].

Q2. The authors highlight that pirfenidone alleviates ARDS inflammation and fibrosis by modulating the TGF-β/Smad signaling pathway. Although they measure TGF-β1, Smad2 and P-Smad2 levels, binding of TGF-β1 to its receptors not only activates and phosphorylates Smad2, but also Smad3 and P-Smad3. It would be important for the authors to evaluate the levels of Smad3 and P-Smad3 to provide a more complete picture of the mechanism of action of pirfenidone in modulating this signaling pathway.

Author answer: We agree with the reviewer’s opinion. According to reviewer’s comment, we newly measured Smad3/p-Smad3 expression results with western blotting. The modified sentence, figure, and figure legend were inserted in manuscript.

  • Following figure were modified in the manuscript

  • Following sentence were inserted in the manuscript

2.4. Expressions of TGF-β, Smad2, and Smad3

The expressions of TGF-β (25 kDa), Smad2 (55-60 kDa), and Smad3 (54 kDa) are shown in Fig. 4. These results indicate that TGF-β, p-Smad2 (60 kDa), and p-Smad3 (54 kDa) expressions were increased upon ARDS induction (P < 0.05). But, treatment of pirfenidone inhibited the expressions of TGF-β, p-Smad2, and p-Smad3 in the ARDS groups (P < 0.05).

Figure 4. Expressions of transforming growth factor (TGF)-β, SMAD family member 2 (Smad2), and Smad3. (A) The relative TGF-β expression in the lung tissue. (B) The relative ratio of phosphorylated Smad2 (p-Smad2) to Smad2 in the lung tissue. (C) The relative ratio of phosphorylated Smad3 (p-Smad3) to Smad3 in the lung tissue.

In the present results, TGF-β1, p-Smad2, and p-Smad3 levels were clearly enhanced by intratracheal LPS and bleomycin injection, and treatment with pirfenidone clearly suppressed LPS/bleomycin-induced TGF-β1, p-Smad2, and p-Smad3 expressions.

Reviewer 3 Report

Comments and Suggestions for Authors

The manuscript “Pirfenidone alleviates inflammation and fibrosis of acute respiratory distress syndrome by modulating the transforming  growth factor-β/Smad signaling pathway by Seung Sook Paik et al. studied the efficiency of pirfenidone on lipopolysaccharide (LPS) and bleomycin-induced ARDS using rats, however, there are several concerns on this manuscript:

1. Line 85However, pirfenidone treatment increased BALF cell number in the ARDS groups, which cell type increased ( monocytes/macrophages, neutrophils/eosinophils or lymphocytes)? What does this mean? Pirfenidone treatment increased ARDS inflammation?

2. The results parts are too simple, the authors are supposed to logically describe why you are going to do this experiment/how you perform your experiment/what results you got/ what the results show... Why do the authors only detect those cytokines (TNF-α, IL-1β, IL-6, and IL-8)? How do the authors directly target TGF-β/Smad2?

3. How the authors perform the experiments is unclear. Which tissues/cells/organs/serum are used is not clear.

4. How many replicates and mice were used in the experiments not included in the legends?

Author Response

Answers to Reviewers’ Comments

Journal name: International Journal of Molecular Sciences
Manuscript ID: ijms-3054832
Type of manuscript: Article
Title: Pirfenidone alleviates inflammation and fibrosis of acute respiratory distress syndrome by modulating the transforming growth factor-β/Smad signaling pathway
Authors: Seung Sook Paik, Jeong Mi Lee, Il-Gyu Ko, Sae Rom Kim, Sung Wook Kang, Jin An, Jin Ah Kim, Dongyon Kim, Lakkyong Hwang, Jun-Jang Jin, Sang-Hoon Kim, Jun-Youl Cha, Cheon Woong Choi*

We sincerely appreciate for your kind advice and comments to our manuscript. We revised the manuscript according to the reviewer’s comments. We added new experimental data, and modifications were expressed in red.

The manuscript “Pirfenidone alleviates inflammation and fibrosis of acute respiratory distress syndrome by modulating the transforming growth factor-β/Smad signaling pathway” by Seung Sook Paik et al. studied the efficiency of pirfenidone on lipopolysaccharide (LPS) and bleomycin-induced ARDS using rats, however, there are several concerns on this manuscript:

Q1. Line 85“However, pirfenidone treatment increased BALF cell number in the ARDS groups”, which cell type increased (monocytes/macrophages, neutrophils/eosinophils or lymphocytes)? What does this mean? Pirfenidone treatment increased ARDS inflammation?

Author answer: First all, thank you for your kind advice and comments. BALF analyzed in this study is total counting. As the reviewer commented, it is difficult to confirm the increase in specific cells. However, in previous studies related to this study, the total number of white blood cells increased due to the induction of lung inflammation, but the number of BALF was found to decrease. However, in previous studies related to this study, the total number of white blood cells increased due to the induction of lung inflammation, but the number of BALF was found to decrease. These results indicate that excessive changes in BALF occurred due to lung damage. It is also evidence that it is difficult to maintain homeostasis within the lung organs.

In the current study, induction of ARDS resulted in a decrease in BALF compared to controls. However, administration of pirfenidone appeared to increase the number of BALFs reduced due to the induction of ARDS. These results mean that it is effective in maintaining normal homeostasis by restoring damaged respiratory organs. Therefore, the increase in BALF following pirfenidone administration is not due to increased inflammation, but is evidence indicating that it is a process of repairing damage.

[Reference]

Ko, I.G.; Hwang, J.J.; Chang, B.S.; Kim, S.H.; Jin, J.J.; Hwang, L.; Kim, C.J.; Choi, C.W. Polydeoxyribonucleotide ameliorates lipopolysaccharide-induced acute lung injury via modulation of the MAPK/NF-κB signaling pathway in rats. Int Immunopharmacol. 2020, 83, 106444. http://doi.org/10.1016/j.intimp.2020.106444

Q2. The results parts are too simple, the authors are supposed to logically describe why you are going to do this experiment/how you perform your experiment/what results you got/ what the results show... Why do the authors only detect those cytokines (TNF-α, IL-1β, IL-6, and IL-8)? How do the authors directly target TGF-β/Smad2?

Author answer: We agree with the reviewer’s opinion. The purpose of this study was to evaluate the therapeutic efficacy of pirfenidone for ARDS. The evaluation of inflammatory cytokines questioned whether the expression of inflammatory cytokines would be maintained even in the fibrotic phase of ARDS. In this study, because ARDS was induced by simultaneous administration of LPS and bleomycin, it was confirmed that the expression of inflammatory cytokines was maintained even on the 21day.

Pirfenidone attenuates pro-fibrotic pathways. AEC damage due to environmental or other factors induces AECs and endothelial cells to augment TGF-β production.

Four key processes are activated: proliferation of fibroblasts, trans differentiation of fibroblasts into myofibroblasts, collagen synthesis and fibronectin production, and excess ECM production. Targets of pirfenidone include TGF-β itself and TGF-β-induced downstream mediators and products such as SMAD3; α-SMA; tenascin-c; fibronectin; collagen type I, II and III; and the collagen-specific chaperone HSP47. Pirfenidone also regulates additional growth factors such as PDGF and bFGF, thereby modulating collagen production. Pirfenidone also inhibits expression of MMP-9, TIMP1, and MMP-2 (yellow boxes) directly or by reducing the synthesis of TGF-β and downstream mediators. Pirfenidone may also reduce TGF-β activation by MMPs (Saleela et al., 2020). In this study, the therapeutic efficacy of pirfenidone in ARDS was demonstrated based on this mechanism of action.

Q3. How the authors perform the experiments is unclear. Which tissues/cells/organs/serum are used is not clear.

Author answer: This study used 40 male adult Sprague-Dawley rats, and after completing all experiments, BALF was collected through the airway, and the right lobe of the lung was collected and used for analysis. This content is well explained in the research methods.

Q4. How many replicates and mice were used in the experiments not included in the legends?

Author answer: Before this experiment, the ARDS model induction method and effective concentration of pirfenidone were selected through two preliminary experiments. And this study used 40 male adult Sprague-Dawley rats. The lung tissue used for analysis was collected from the right lobe, which was divided into equal parts and used for ELISA, western blotting, and histological evaluation.

Round 2

Reviewer 1 Report

Comments and Suggestions for Authors

Accept in current form 

Author Response

Reviewer -1

Q1. Accept in current form

 Author answer: Thank you for your kind advice and decisions.

Reviewer 3 Report

Comments and Suggestions for Authors

The authors did not improve the article. 

1. It is easy to confirm the specific cell increase by different cell markers and perform flow cytometry.

Author Response

Reviewer -3

The authors did not improve the article.

Q1. It is easy to confirm the specific cell increase by different cell markers and perform flow cytometry.

Author answer: First all, thank you for your kind advice and comments. Although we must conduct experiments using flow cytometry according to the reviewer's comment, we are not in a position to conduct experiments using flow cytometry. For that reason, different cells were counted using a different method. According to reviewer’s comment, the modified sentence was inserted in method part. In addition, new figure was added in supplementary data part.

  • Following sentence were added to the Method part

4.4. BALF cell counting

BALF cell counting was done in the following way [3,36]. BALF cell suspension was diluted at 1:20 with trypan blue. After loading into the hemocytometer chamber and set-tling the cells, cells were counted in four corner squares. Cell differential counts were per-formed on cytospin preparations (Shadon, Pittsburgh, PA, USA), and the cells were stained with Diff-Quick (Fisher Scientific; NJ, USA). Total leukocytes were counted using light microscopy (Olympus, Tokyo, Japan).

Furthermore, differential counts of eosinophils, neutrophils, and monocytes were determined on cytospin smears of BAL samples from individual rat stained with Diff-Quick and identified by standard morphological criteria after counting cells. The results of differential cell counts (eosinophils, neutrophils, and monocytes) are represented in supplementary result 1.

  • Following figure were inserted in the supplementary result part

Supplementary Figure 1. Cell differ-counting of bronchoalveolar lavage fluid. * shows P < 0.05 compared to the control group. # shows P < 0.05 compared to the ARDS-inducing group. CON, control group; ARDS, acute respiratory distress syndrome-inducing group; ARDS-P100, acute respiratory distress syndrome-inducing and 100 mg/kg pirfenidone-treating group; ARDS-P200, acute respiratory distress syndrome-inducing and 200 mg/kg pirfenidone-treating group.

Round 3

Reviewer 3 Report

Comments and Suggestions for Authors

Please revise the results part according to my previous suggestions.

Author Response

Answers to Reviewers’ Comments

Journal name: International Journal of Molecular Sciences
Manuscript ID: ijms-3054832
Type of manuscript: Article
Title: Pirfenidone alleviates inflammation and fibrosis of acute respiratory distress syndrome by modulating the transforming growth factor-β/Smad signaling pathway
Authors: Seung Sook Paik, Jeong Mi Lee, Il-Gyu Ko, Sae Rom Kim, Sung Wook Kang, Jin An, Jin Ah Kim, Dongyon Kim, Lakkyong Hwang, Jun-Jang Jin, Sang-Hoon Kim, Jun-Youl Cha, Cheon Woong Choi*

We sincerely appreciate for your kind advice and comments to our manuscript. We revised the manuscript according to the reviewer’s comments. We added new experimental data, and modifications were expressed in red.

Reviewer -3

Q1. Please revise the results part according to my previous suggestions.

Author answer: Thank you for your kind advice and comments. According to reviewer’s comment, the modified sentence was inserted in result part.

  • Following sentence were added to the result part

2.1. Cell counting of BALF

The results of BALF cell count are represented in Fig. 1. BALF cell number was de-creased upon ARDS induction (P < 0.05). However, pirfenidone treatment increased BALF cell number in the ARDS groups (P < 0.05). The results of the differential cell counts (eosinophils, neutrophils, and monocytes) are presented in supplementary Fig. 1. Differential cell counts showed that the numbers of esosinphils and neutrophils increased with ARDS induction and pirfenidone treatment. However, the number of monocytes decreased in the ARDS group, whereas treatment with prifenidone increased monocyte numbers.
